# Asthma severity as a contributing factor to cancer incidence: A cohort study

**Laila Salameh**[1], **Bassam Mahboub**[2], **Amar Khamis**[3], **Mouza Alsharhan**[2], **Syed Hammad Tirmazy**[2], **Youssef Dairi**[2], **Qutayba Hamid**[1,4], **Rifat Hamoudi** [1,5‡], **Saba Al Heialy** [3,4‡]*

1 College of Medicine, University of Sharjah, Sharjah, United Arab Emirates, 2 Dubai Health Authority, Dubai, United Arab Emirates, 3 Mohammed bin Rashid University of Medicine and Health Sciences, Dubai, United Arab Emirates, 4 Meakins-Christie Laboratories, Research Institute of the McGill University Healthy Center, College of Medicine, Montreal, Quebec, Canada, 5 Division of Surgery and Interventional Science, UCL, London, United Kingdom

‡ These authors are joint senior authors on this work.
* saba.alheialy@mbru.ac.ae

## Abstract

### Background

A putative link between asthma and asthma severity with the occurrence of cancer has been suggested but has not been fully investigated. The objective of this study is to assess the incidence of all types of cancer in a cohort of asthmatic patients.

### Methods and findings

A single center cohort retrospective study was conducted to investigate the role of asthma as a potential risk factor for various cancers. Participants were followed for a period of 9 years from 01/01/2010 to 30/12/2018 and cancer incidence and its determinants were collected in asthmatic patients and controls from the same population source but without any respiratory disease. Overall, 2,027 asthma patients and 1,637 controls were followed up for an average of 9 years. The statistical analysis showed that 2% of asthma patients were diagnosed with various cancers, resulting in an incidence rate of cancer of 383.02 per 100,000 persons per year which is significantly higher than the 139.01 per 100,000 persons per year observed in matched controls (*p*-value < 0.001). The top four cancers reported among asthmatics were breast, colon, lung and prostate cancer. Lung cancer in asthmatics had the longest diagnosis period with a mean of 36.6 years compared to the shortest with prostate cancer with 16.5 years.

### Conclusions

This study shows that asthma patients are at increased risk of different types of cancers with asthma severity and goiter as the main factors that may increase the risk of developing cancers among asthmatic patients.

**Funding:** SA the senior author received fund by Al-Jalila Foundation (Grant code: AJF2018113). https://www.aljalilafoundation.ae.

**Competing interests:** The authors declare no competing interests.

## Introduction

Worldwide incidence of cancer is on the rise. In 2018, 18.1 million new cases were reported with 9.6 million deaths. One in 5 males and one in 6 females worldwide develop cancer during their lifetime, and one in 8 males and one in 11 females die from the disease [1].

The increase of cancer incidence has prompted research on the factors that play a role in promoting or protecting against oncogenesis [2–4]. One of the main areas of interest has been the relationship between cancer and chronic inflammatory diseases. Recent evidence strengthened the concept of an association of inflammation and cancer [5–7] and nowadays it is generally accepted that up to 25% of human malignancies are related to chronic inflammation and to viral or bacterial infections [8]. One of the common conditions that affects the lung, where inflammation and the immune response are heavily implicated, is asthma. Global Initiative for Asthma (GINA) describes asthma as a heterogeneous disease, usually characterized by chronic airway inflammation with signs of wheezing, cough, dyspnea and chest tightness [9]. Asthma severity varies between the individuals and within the individuals overtime [9]. In mild-moderate asthma, using bronchodilators, inhaled corticosteroids (IC) and asthma management plan is the basis of effective treatment [10]. Moreover, there are 3–10% [10–12] of asthmatic patients who fail to respond to standard therapy and are categorized as severe asthmatics. Therefore, asthma severity is defined as "asthma which requires maximum controller therapy to prevent a patient from becoming uncontrolled or which, despite high dose therapy remains uncontrolled" [12]. Patients with severe asthma suffer from significant difficulties in daily living and decrease in physical activity and work productivity [12, 13]. Furthermore, they face increased comorbidity burden [12].

It is believed that allergies and asthma, inflammatory conditions commonly characterized by immunoglobulin E (IgE)-mediated atopic reactions [14], may either decrease cancer risk via increase in immunosurveillance or may increase risk due to persistent immune stimulation [15]. Recent studies demonstrated that the association between history of allergic disorders and asthma, and the risk of cancer varied by cancer types [16–19]. A protective effect of atopic diseases against pancreatic cancer [20, 21] has been shown consistently in case-control studies but not in cohort studies. Most studies on atopic diseases and non-Hodgkin lymphoma or colorectal cancer reported an inverse association. Inversion association between cancer and asthma have been reported in case-control studies of glioma [22, 23], and leukemia [24, 25]. In addition, controversial results have been reported in several studies in lung and breast cancer [15, 18, 19, 26–30]. However, the studies showing no association are based largely on qualitative data and rely on surveys. In addition, misdiagnosis of non-asthmatic conditions as uncontrolled asthma has been reported to be as high as 12–30% [12]. This might be because asthma symptoms may be similar to symptoms in other lung diseases such as chronic obstructive pulmonary disease (COPD), emphysema and chronic bronchitis. Interestingly, similar symptoms to severe asthma can also be seen in patients with early symptoms of lung cancer [31]. Asthmatic patients and even their treating physicians may delay diagnosis for these symptoms and attribute them to uncontrolled asthma. In addition, non-smoker patients with asthma are generally excluded from high-risk group eligible for lung cancer screening [32]. Most patients delay seeking medical care until ominous symptoms like hemoptysis, pain, pleural effusion or distant metastasis develop, however by then radiological studies would most of the time detect late-stage lung cancer that is beyond surgical cure.

Therefore, using patient data records, rather than surveys, and stratifying asthma severity is highly needed to address these limitations and to more fully evaluate associations between asthma and specific cancer sites. The aim of this study is to investigate the role of asthma as a potential risk factor for cancer using clinical records of patients from the main referral hospital

for respiratory disease in the UAE, and if present to identify the factors contributing to this association.

## Materials and methods

### Ethical consideration

All patient records were collected from the computerized information health system in Dubai Health Authority. All records in the outpatient and inpatient register, containing a hospital-discharge diagnosis of asthma. The data accessed were from 24/9/2019–30/4/2020. All patients were consented. Patient records were fully anonymized prior to access.

The study protocol was reviewed and approved by Dubai Scientific Research Ethical Committee (DSREC) Dubai Health Authority; the ethical approval number of the study is DSREC-SR-03L2019_01.

### Study population

**Asthmatic cases.** The details of the study cohort were derived retrospectively from the medical records department in Rashid Hospital and Dubai Hospital which are part of the Dubai Healthcare Authority. UAE patients (local Emirati and expatriates) diagnosed with asthma follow-up at Rashid Hospital whereas patients with cancer follow-up in Dubai Hospital. Yearly, around 600–700 patients examined in Rashid Hospital are diagnosed with asthma as primary diagnosis (ICD-10 code). Each patient has a unique health card number, which is used to identify each case and ensure no duplication in the records. The records of patients diagnosed with asthma between January 1, 2010 and December 31, 2018 were retrieved. The total number of asthmatic patients identified was 2027 (prevalent cases). Patients were eligible for inclusion if they were 20 years old or older, were diagnosed with asthma as primary diagnosis (ICD-10 code) at time of discharge as in-patients or after their visit in the outpatient clinic and had more than one visit for at least one year in the hospital before 2010. Power calculation based on similar studies but a different population such as Boffetta P *et al.* (20), showed that a minimum of 1345 asthmatic patients is needed to identify sufficient cohort of cancers to determine whether an association between asthma and cancer exist. The parameters for the power calculation were chosen as follows: significance level = 0.05, power = 90%, the standard deviation = 16 and effect size = 2 were obtained from Boffetta P *et al.* which carried out the study on Swedish population. Therefore, the final total number of 1886 identified from the records (Fig 1) is sufficient for the study. For the classification of asthma severity, we followed GINA guidelines [9] (S1 Table) where mild asthma is described as patients prescribed step 1–2 treatment, whereas moderate asthma is defined as patients prescribed treatment step 3–4 without history of asthma exacerbation or one in the last year of participation of the study. Severe asthma is defined as patients prescribed treatment step 4–5 with history of at least 2 exacerbations in the last year and/or a history of hospital admission for acute severe asthma in the last year of participation in the study.

In order to identify cancer patients (incidence cases) within the asthma cohort, all cohort linked to cancer registry in Dubai Health Authority were classified according to the ICD-10 code. The clinical records were retrieved, and cancer diagnosis was reviewed. The analysis was restricted to include cases diagnosed with cancer during the study period and excluded eighteen cases diagnosed with cancer before January 1, 2010. The total cohort, which includes all patients who followed up from beginning of 2010 until end of 2018, was 1868 patients with 38 patients diagnosed with different types of cancer.

**Control cases.** Matched controls against age and gender with persons attending the same hospitals but with other diseases were used (n = 2262). The inclusion criteria for controls were 20 years old or older, no past or present diagnosis of asthma and other pulmonary diseases

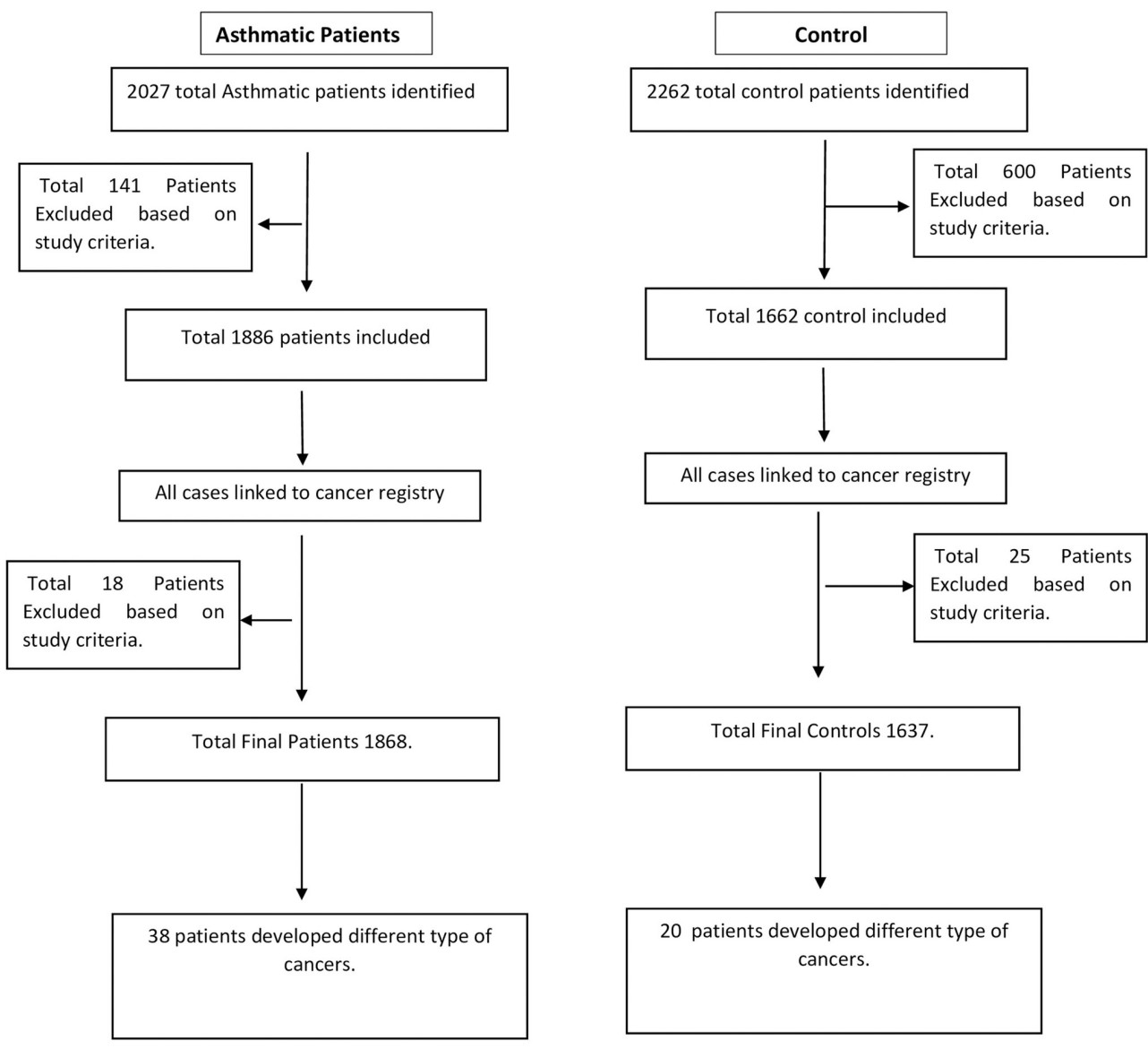

**Fig 1. The flow chart of the study methodology.**

and other symptoms of allergic diseases such as nasal and skin symptoms. Control patients were free of malignancy at the time of recruitment between January 1, 2010 and December 31, 2018. A total of 600 patients were excluded according to the criteria and 1662 patients were identified. In order to identify cancer patients within the control, all records linked to cancer registry in Dubai Health Authority were classified according to the ICD-10 code. The clinical records were retrieved, and cancer diagnosis was reviewed. The analysis was restricted to include cases diagnosed with cancer during the study period and excluded 25 cases diagnosed with cancer before January 1, 2010. The total control cases were 1637 (Fig 1).

## Statistical analysis

Parametric tests including descriptive analysis, incidence analysis was used for quantitative variables. Fisher exact test, the Chi-squared test and Cox logistic regression were applied for

association analysis and comparison of categorical variables. Kaplan-Meier survival analysis was carried out to determine the progression of patients from asthma to cancer and log rank was used to determine whether the variables compared are significant. Two-sided p values <0.05 were considered to be statistically significant. Statistical package SPSS (version 24) was used for statistical analyses.

### Univariate analysis

In order to identify the individual risk variables, univariate analysis on each independent variable was applied to the asthmatic cohort. The independent variables consisted of demographic variables, including age, sex, nationality, smoking status, asthma severity as well as comorbid conditions, including, cardiovascular diseases, type 2 diabetes mellitus (T2DM), hypertension (HTN), obstructive sleep apnea (OSA), allergic rhinitis (AR), gastroesophageal reflux disease (GERD), urticaria, hyperlipidemia, obesity, hypothyroidism, goiter and auto-immune diseases including, irritable bowel disease (IBD), rheumatoid arthritis and Hashimoto's thyroiditis. All selected variables were tested for co-linearity to avoid any strong correlation between the variables. The presence of co-linearity was examined by the evaluation of variance inflation factors and magnitude of standard errors. Variables with more than 30% missing values were not included in the analysis.

### Multivariate analysis

The variables that were significant from the univariate analysis were incorporated into multivariate Cox regression model. The multivariate Cox regression was used to determine the degree of interaction between the different variables obtained from the patients' records. The aim of this is to identify which set of variables are key in driving the association between cancer and asthma and their degree of interaction.

In addition, progression from asthma to lung cancer was studied by organizing the data in chronological order of patients who developed asthma and later on progressed to lung cancer. The data was organized according to age at diagnosis of asthma, age at diagnosis of lung cancer, stage of the lung cancer at diagnosis, duration between the diagnosis of asthma and lung cancer and number of asthma attacks per year over the period of their follow up.

### Results

Descriptive statistics using patient cohort identified in this study are shown in Table 1.

Table 1 shows that the total asthmatic patients in the cohort were 1868 patients with a mean age of the cohort study of 50 ±16.4 years compared to 52 ±13.8 years for control patients. Females comprised 68% of the asthmatic cohort and Emirati nationals were 87% compared to 69% and 86% respectively in the control patients. 13% of asthmatic patients had a history of smoking compared to 14% in control patients. Table 2 shows the top four cancers in asthmatic patients compared to control group. Detailed analysis of the asthmatic cohort of all comorbidities reported including the asthma severity (S2 Table) shows that the affected individuals with the selected co-morbidities are much less than the unaffected individuals, resulting in a difficulty in identifying epidemiological patterns in the data. However, the small number of affected individuals does show that there is a link between autoimmune, allergic disease, thyroid and asthma.

In terms of cancers, we identified that out of 1868 cases of asthma followed for 9 years, 38 (2%) patients were diagnosed with cancer compared to 20 (1.2%) control patients; this makes the incidence rate of cancer among asthmatics 383.02 per 100,000 person per year compared

**Table 1. Characteristics and comorbidities for asthmatic and control patients during 2010–2018.**

| Basic Demographic | Asthmatic cases n = (1868) | Controls n = (1637) | P value |
|---|---|---|---|
| Gender (F) *N* (%) | 1264(68) | 1128(69) | 0.226 |
| Age (in years) M ± SD | 50 ±16.4 | 52 ± 13.8 | 0.294 |
| Nationality (Emirati) *N* (%) | 1627(87) | 1408(86) | 0.186 |
| Smoking History (Smoker) *N* (%) | 241(13) | 225 (14) | 0.247 |
| Developed cancer *N* (%) | 38(2) | 20 (1.2) | 0.039 |

**Table 2. Number and incidence of cancer per 1 million person-months among asthmatic cohort compared to controls.**

| Asthmatic Cases who developed Cancer N (%) | | Asthmatic (Incidence of cancer per 1 million per person-months) | Control Cases who developed Cancer N (%) | | Controls (Incidence of cancer per 1 million per person- months) |
|---|---|---|---|---|---|
| Breast Cancer | 11 (0.6) | 93.75 | Breast Cancer | 5 (0.3) | 30.8 |
| Colon Cancer | 8 (0.4) | 68.18 | Brain Tumor | 2 (0.1) | 12.32 |
| Lung Cancer | 6 (0.3) | 51.13 | Papillary Thyroid Cancer | 2 (0.1) | 12.32 |
| Prostate Cancer | 2 (0.1) | 17.04 | Prostate Cancer | 2 (0.1) | 12.32 |

to 139.01 per 100,000 person per year. There is a significantly higher incidence among asthma patients compared to control patients (p-value < 0.001).

The incidence rate presented in Table 3 shows that the leading cancers among the asthmatic cohort are: breast cancer (n = 11) with 93.8 per 1 million person-months; colon cancer (n = 8) with 68.2 per 1 million person per month; lung cancer (n = 6) with 51.1 per 1 million person per month and prostate cancer with 17 per 1 million persons per months. Whereas the control cohort showed the following top four cancers: breast cancer (n = 5) with 30.8 per 1 million person per months; brain tumor (n = 2) with 12.32 per 1 million person per months; papillary thyroid cancer (n = 2) with 12.32 per 1 million person per months; prostate cancer (n = 2) with

**Table 3. Association between cancer among asthma patients and other co-morbidities.**

| Comorbidities | Asthmatic patient free from cancer n(%) | Asthmatic patient with cancer n(%) | OR (95%CI) | *p*-value |
|---|---|---|---|---|
| Goiter | 16(1) | 4(11) | 13.34(4.24–41.91) | 0.001 |
| HTN | 412(23) | 17(45) | 2.79(1.146–5.33) | 0.002 |
| Hyperlipidemia | 269(15) | 11(29) | 2.36(1.16–4.82) | 0.019 |
| Cardiovascular | 96(5) | 5(13) | 2.74(1.05–7.17) | 0.051 |
| AR | 677(37) | 10(26) | 0.61(0.30–1.26) | 0.117 |
| DM | 455(25) | 13(34) | 1.57(0.80–3.10) | 0.13 |
| IBD | 6(0.3) | 1(3) | 8.22(0.97–69.96) | 0.134 |
| Autoimmune Disease | 91(5) | 3(8) | 1.64(0.49–5.43) | 0.298 |
| Hashimoto's Thyroiditis | 17(1) | 1(3) | 2.88(0.37–22.23) | 0.31 |
| Urticaria | 19(1) | 1(3) | 2.58(0.34–19.75) | 0.338 |
| OSA | 119(7) | 3(8) | 1.23(0.37–4.07) | 0.457 |
| GERD | 87(5) | 2(5) | 1.11(0.26–4.70) | 0.549 |
| Hypothyroidism | 95(5) | 2(5) | 1.02(0.24–4.28) | 0.596 |
| Rheumatoid Arthritis | 62(3) | 1(3) | 0.77(0.10–571) | 0.631 |

12.32 per 1 million person per months. The data show that the top two cancers (breast and colon) are driven by hormonal status followed by lung cancer. Therefore, lung cancer is the top disease outside the hormonal-driven cancers.

## Univariate analysis of asthma variables

The univariate analysis between asthma patients with cancer and asthma patients without cancer (S3 Table) in relation to the demographic data (gender, nationality, history of smoking) and the severity of asthma was assessed and Q-Q plot was used to check the normality distribution. The results show that there was no indication of increased risk of cancer in relation to the following variables: smoking ($p = 0.21$), gender ($p = 0.462$) and nationality ($p = 0.21$). However, severity of asthma in patients with cancer was statistically significant ($p<0.001$) with significant differences between mild asthmatic compared to moderate and severe asthma ($p<0.001$) while there were no significant changes between moderate and severe asthma ($p<0.082$). Interestingly, 82% of asthmatic patients who didn't smoke developed cancer.

Table 3 shows the univariate analysis between asthma patients with cancer and asthma patients without cancer in relation to other co-morbidities. Results show that patients with HTN had 2.79 (95%CI; 1.146–5.33) times more chance of developing cancer compared to patients without HTN. Similarly, patients with hyperlipidemia had 2.36 (95%CI; 1.16–4.82) times more chance of developing cancer compared to patients without hyperlipidemia. Interestingly patients with goiter had 13.34 (95%CI; 4.24–41.90) times more chance of developing cancer compared to patients without goiter.

## Multivariate analysis using Cox-Model

Table 4 shows multivariate analysis using Cox-Model. In this model, we use the variables of cancer (dichotomous variables) as dependent variables and all the variables that are significant using univariate analysis (age, asthma severity, HTN, hyperlipidemia and goiter).

A multivariate Cox proportional hazards regression analysis adjusted for age, asthma severity, hypertension, hyperlipidemia and goiter revealed that among the cohort asthma severity ($p< 001$; 95% CI, 2.005–5.389) and goiter ($p = 0.10$; 95% CI, 0.018–0.576) were a risk for developing cancer. The data show that severity and goiter remain significant in the Cox model analysis suggesting that there might be a link between goiter and asthmatic disease.

## Effect of asthma persistence/progression on the onset of cancer

Fig 2 shows the effect of asthma persistence or progression on the onset of cancer. The data shows that there are significant differences ($p < 0.001$) between (mild- moderate) asthmatic and (mild-severe) asthmatic progression towards developing cancer while between (moderate-severe) asthmatic was marginal ($p = 0.04$).

**Table 4. Multivariate analysis using Cox-Model of the significant variables identified from the univariate analysis of the asthmatic cohort with and without cancer.**

| Variables | B | Sig. | Exp(B) | 95.0% CI for Exp(B) | |
|---|---|---|---|---|---|
| | | | | Lower | Upper |
| Age Group | 1.133 | 0.015 | 3.104 | 1.241 | 7.763 |
| Asthma Severity | 1.074 | 0.000 | 2.929 | 1.784 | 4.807 |
| HTN | 0.242 | 0.491 | 1.273 | 0.641 | 2.531 |
| Hyperlipidemia | -0.072 | 0.848 | 0.930 | 0.444 | 1.948 |
| Goiter | 2.280 | 0.000 | 9.781 | 3.260 | 29.348 |

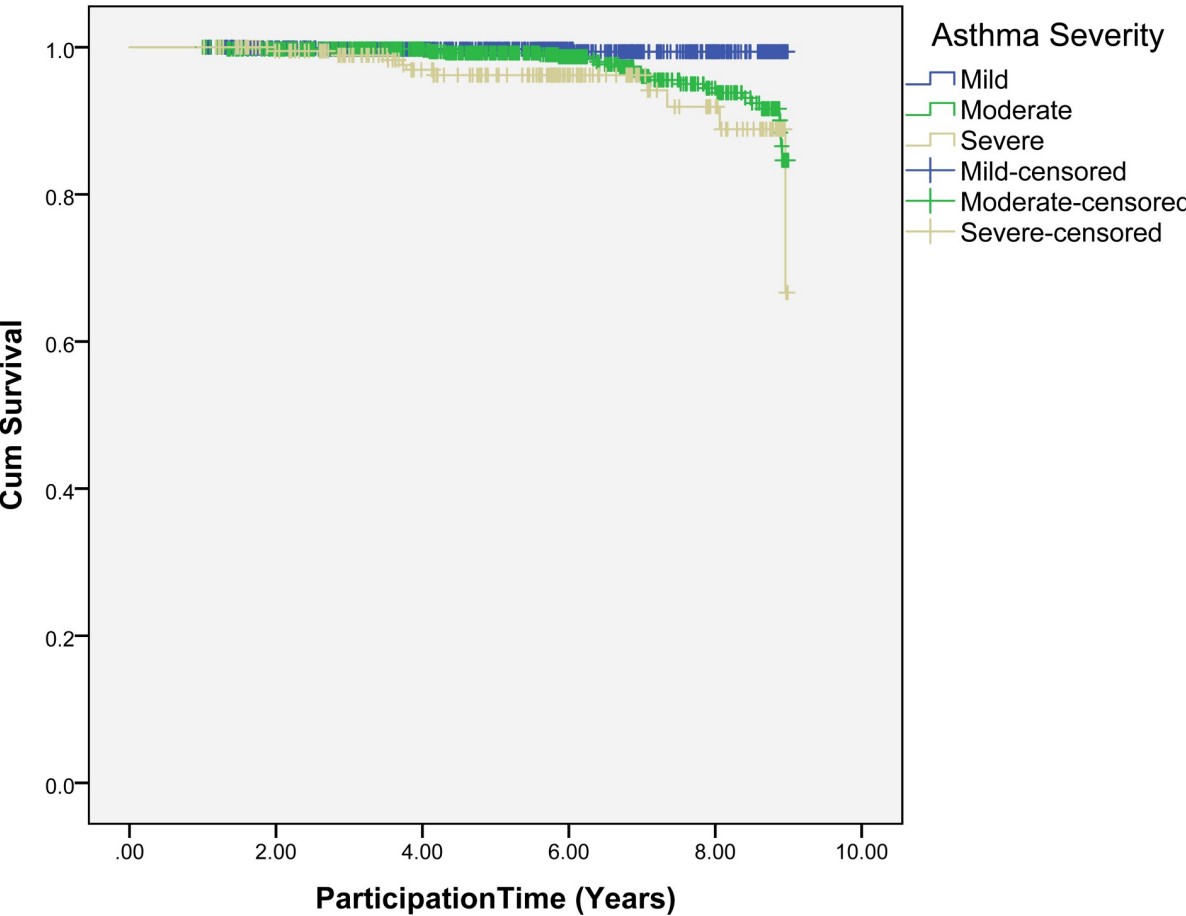

**Fig 2. Speed of the occurrence of cancer among asthma patients associated with severity of asthma.**

Fig 3 describes the average number of years for cancer diagnosis according to the top four cancers identified in Table 2. Lung cancer diagnosis was revealed to have the longest diagnosis period with an average of 36.6 years compared to prostate cancer with 16.5 years. This indicates that asthma might play a role in developing cancer through low level perturbation of the tissue.

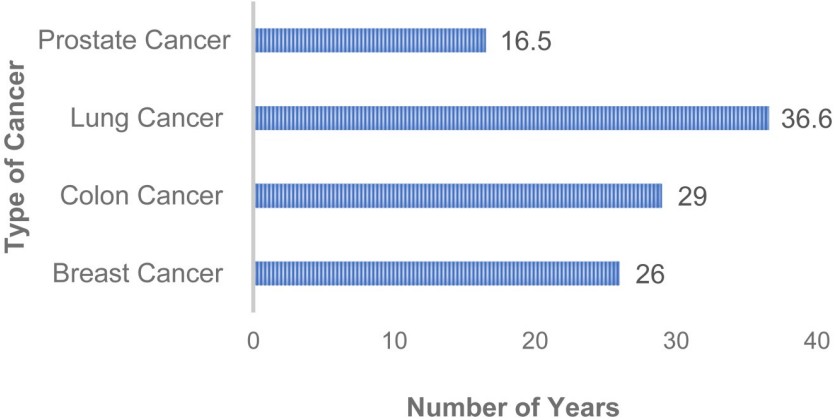

**Fig 3. Average of years between asthma and cancer diagnosis.**

50% of lung cancer with asthma patients are diagnosed with stage 4 lung cancer (S1 Fig). This is typical in lung cancer as most of the cases in lung cancer tend be in the later stages of the disease. Interestingly, a similar number is found for prostate cancer where 50% of patients have early prostate cancer patients compared to 50% with aggressive prostate cancer. This suggests that inflammation and chronic inflammation play a role in eventually establishing cancer since prostate cancer is an indolent disease characterized by long term chronic inflammation which eventually develops into cancer after a lengthy period. On the other hand, colorectal and breast cancers are more aggressive and perhaps asthma indirectly contribute to its progression. Comparison between alive and dead asthmatic cases from the top four type of cancer in asthmatic cohort (S2 Fig) was assessed. Data shows that prostate cancer has the highest survival because it is indolent whereas lung cancer has the lowest survival because it is generally diagnosed at late stage.

Later stages of cancer show poor survival of around 57 months whereas stages 2 to 3 show longer survival of asthmatic patients whereas in control patients the survival is 80 months (Fig 4). This indicates that asthma contributes to poorer survival of cancer patients. The outcome from this is supported by the finding in S2 Fig.

Since we have shown that the average years for lung cancer diagnosis among the asthma cohort was high, we were interested to further examine this cohort of patients. Fig 5 shows the details of the follow-up of the 6 cases of lung cancer with asthma patients. Cases 1 and 4 show patients were diagnosed with asthma at the ages of 46 and 41 years respectively. These patients were then diagnosed with stage 4 lung cancer at ages 80 and 79 years respectively. Interestingly, these patients had recurrent admissions to the hospital and were on long-term use of inhaled steroid therapy for asthma. Fig 4 shows that moderate asthma lead to stage II lung cancer but severe asthma lead to the more aggressive stage 3 and 4 lung cancer. This is important as it indicates that asthma plays a contributing factor to lung cancer and may be used as an early predictor of the disease. Fig 5 shows that once cancer has been established, the follow-up after cancer diagnosis is variable and probably depends on the severity of asthma.

## Discussion

To date, asthma and lung cancer are viewed as independent diseases although they affect the same organ and share common clinical manifestations. In this study, we have shown that asthma may, in some cases, be a precursor to lung cancer where asthma is an inflammatory disease. Infections or injuries result in the disruption of tissue homeostasis, triggering an inflammatory response initiated by the innate immune system followed by a coordinated response by the adaptive immune system. Various mechanisms can be used by the innate immune cells to initiate inflammation including the release of cytokines, chemokines, matrix-remodeling proteases, and reactive species. However, imprecise regulation of these mechanisms of inflammation can adversely result in chronic inflammation and pro-tumor microenvironment [33]. The mechanisms used by inflammatory conditions to promote tumor growth are numerous, extremely complex and involve lengthy processes. Asthma is a heterogeneous disorder of the conducting airways involving chronic airway inflammation, declining airway function and tissue remodeling [34]. Asthma is thought to arise from the complex interplay of genetic susceptibility and environmental influences, such as timing and dose of allergen and co-exposure to infection [35].

To our knowledge this is the first study in the Middle East examining the association between asthma and risk of cancer, which is important because of the different lifestyles of the analyzed patients, which influences the prevalence of asthma and cancers. In this study, association was observed between history of asthma and risk of various cancers. In our study, asthma

(A)

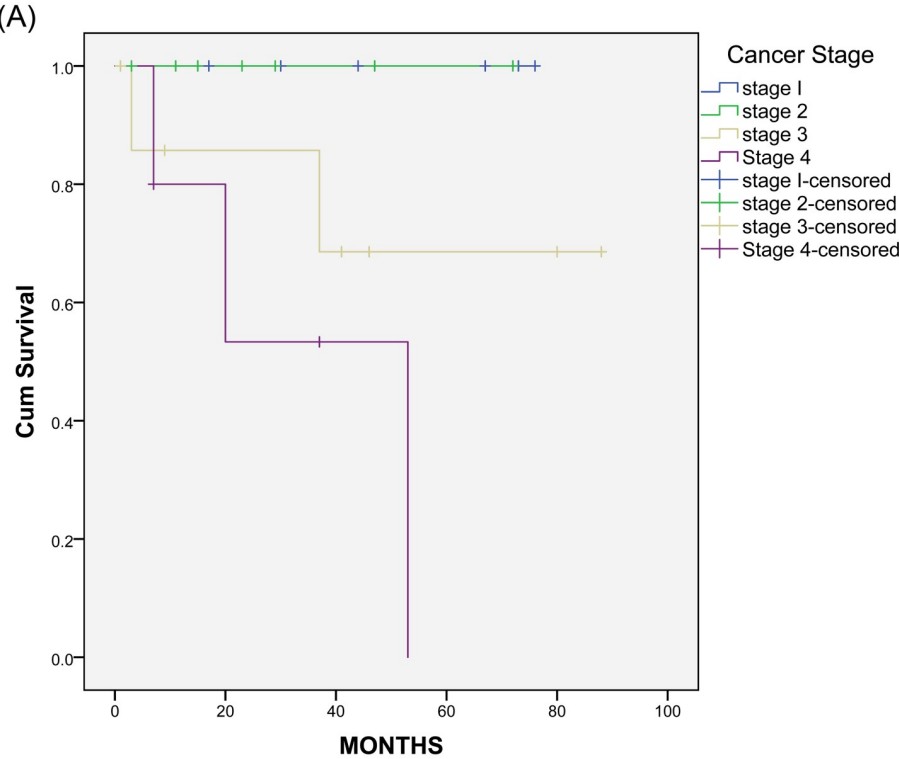

(B)

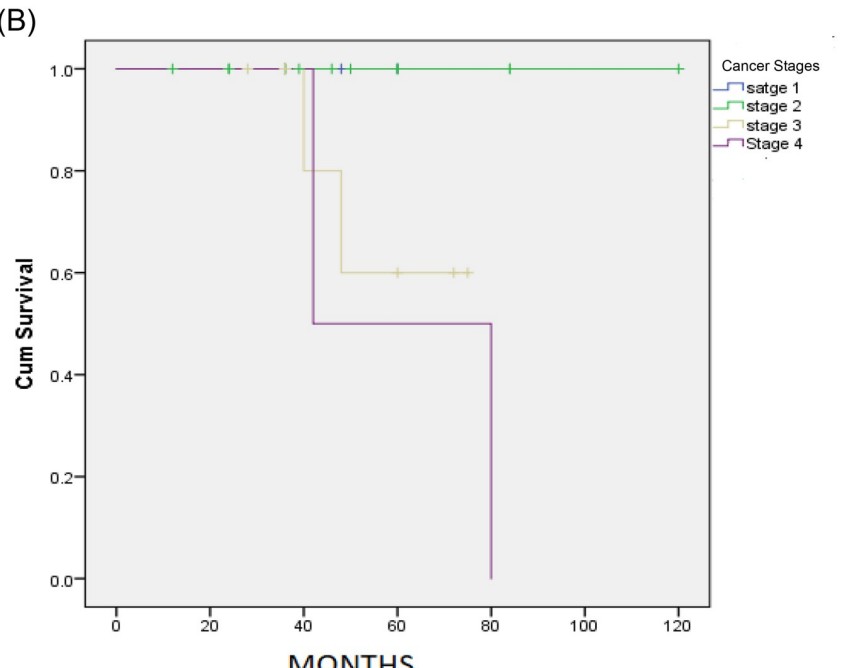

**Fig 4. Kaplan-Meier survival curve among asthmatic and control patients.** (A) Kaplan-Meier survival curve showing the speed of the occurrence of death amongst asthma patients who had cancer at different stages. There was no significance in death progression among patients who in stage 1,2, and 3 (P = *0.2)* but there were significant between stage 1,2 and 4 (*P = 0.01)*. (B) Kaplan-Meier survival curve showing the speed of the occurrence of death amongst control patients with cancer at different stages.

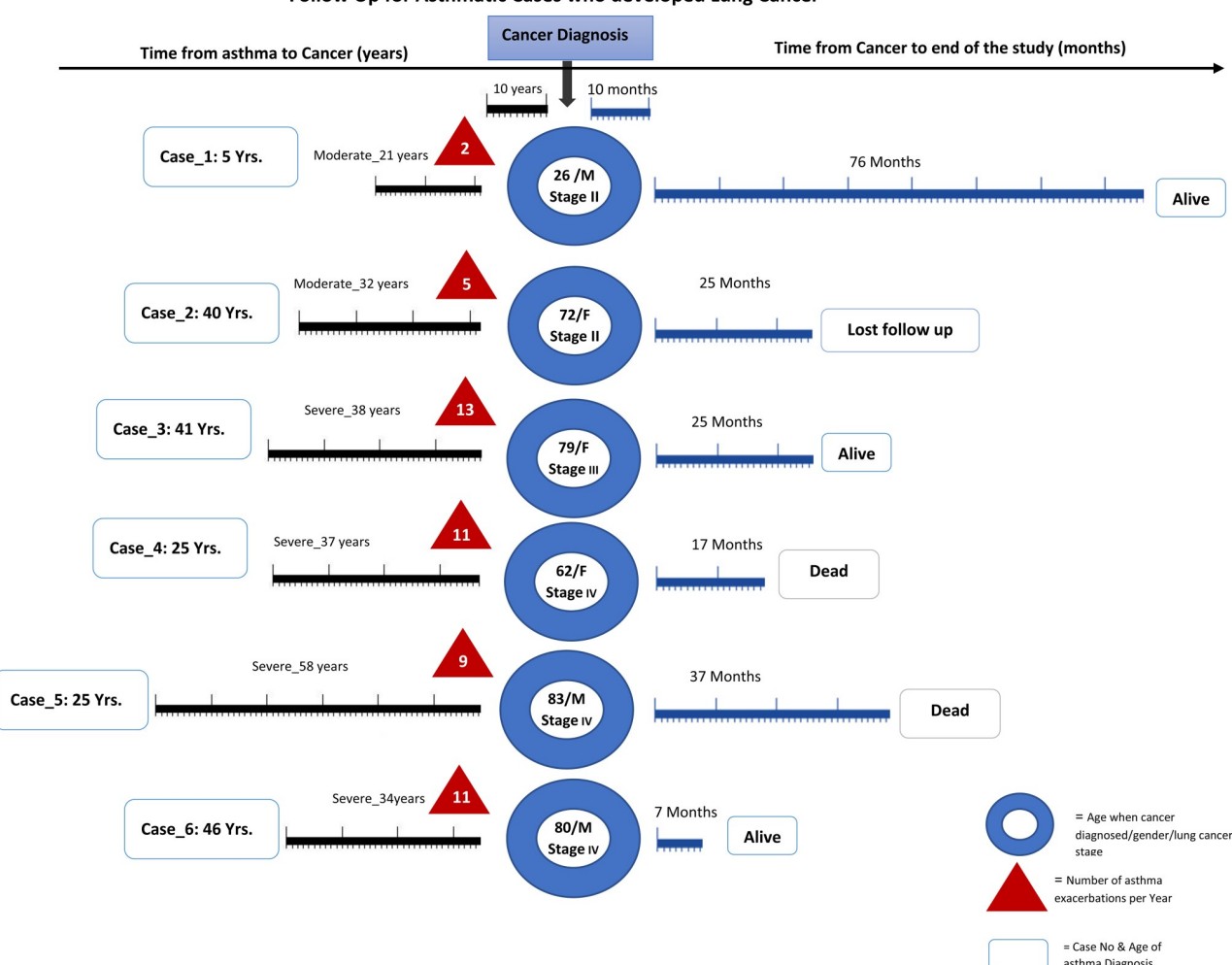

**Fig 5. Follow-up for asthmatic cases who developed lung cancer.**

was associated mainly with increased risk of breast, colon, lung, and prostate cancers which is consistent with data from other publications [17, 19, 26–28, 36, 37]. This is of interest because the hypothesized mechanisms by which allergies and asthma are suspected to affect cancer risk is expected to apply regardless of site [15].

We confirmed through the univariate analysis, that gender and smoking habits do not associate with the risk of cancer. Interestingly, asthma severity shows that there is an association between cancer and asthma. These results suggested that asthma might be an independent risk factor for cancer.

A noteworthy finding in our study was that severe asthmatic patients with goiter were significantly associated with higher speed of the occurrence of cancer cases compared to mild and moderate asthmatic patients. Goiter is an abnormal enlargement of thyroid gland. The location of the gland makes this enlargement important because it can compress the trachea [38]. The relationship between asthma and thyroid problems has been documented in many studies [39–44] and correction of hypothyroidism causes worsening of symptoms and difficulties in controlling asthma [44]. Goiter is due to an imbalance in the regulation of thyroid hormones brought on by iodine deficiency, rare inherited disorders of the thyroid gland or

autoimmune diseases such as Hashimoto thyroiditis and Grave's disease. The loss of immuno-logical tolerance to thyroid antigens leads to the generation of components of the cellular and humoral arms of the adaptive immune response. The infiltration of lymphocytes (more specifi-cally T lymphocytes) and/or the generation of pathogenic autoantibodies causes damage to the thyroid and modifies the response of the thyroid gland [45]. There is reduced responses to bronchodilator therapy with an inverse relationship between levels of thyroid function and air-way beta adrenergic responsiveness [41]. Taken together, this may explain the increase in severity of asthma among this group due to goiter.

In our study, this was established through careful assessment of asthmatic cases by doing full review for their past medical history with different comorbidities and level of asthma severity and this is the first study in the region. In addition, the follow-up analysis of the patients identified a subset of 6 patients (2%) that were diagnosed with both asthma and lung cancer and the survival analysis showed that asthma can be a contributing factor to cancer progression and especially lung cancer. This provides solid evidence of the association between asthma and lung cancer and 2% is in line with the data published by Boffetta et al [26]. The main limitation of our study is that it is from one center, but it is the largest center in the UAE.

Thus, the most prominent finding in our study is the positive association between asthma severity and progression of developing different cancer types. Ji and colleagues showed in their study that the asthmatic patients with multiple hospital admissions had a high risk of develop-ing different types of cancer, particularly for stomach (SIR 1.70) and colon cancers (SIR 1.99) [36].

In summary, our results show real association between asthma and lung cancer by identify-ing a subset of patients that were diagnosed with both diseases. In addition, our study showed through epidemiological analysis that asthma can be a contributing factor to carcinogenesis and its progression in various tissues and specifically the lung, warranting further studies to understand the molecular basis of the link between asthma and lung cancer.

## Supporting information

**S1 Table. GINA guidelines step-up medications.**
(TIF)

**S2 Table. Detailed comorbidities for asthmatic patients during 2010–2018 period.**
(TIF)

**S3 Table. Association between cancer, demographic data and asthma severity among the cohort.**
(TIF)

**S1 Fig. Percentage of cancer stages for the top four cancers in the cohort.**
(TIF)

**S2 Fig. Percentage of death among the top four cancers in asthma cohort.**
(TIF)

**S1 File.**
(SAV)

**S2 File.**
(SAV)

## Acknowledgments

The authors would like to thank Dubai Scientific Research Ethical Committee (DSREC) Dubai Health Authority, Rashid hospital, Dubai hospital for their support in this study.

## Author Contributions

**Conceptualization:** Laila Salameh, Bassam Mahboub, Youssef Dairi, Qutayba Hamid, Rifat Hamoudi, Saba Al Heialy.

**Data curation:** Laila Salameh, Bassam Mahboub, Amar Khamis, Mouza Alsharhan, Syed Hammad Tirmazy, Youssef Dairi, Rifat Hamoudi.

**Formal analysis:** Laila Salameh, Amar Khamis.

**Investigation:** Laila Salameh, Mouza Alsharhan, Syed Hammad Tirmazy, Youssef Dairi, Rifat Hamoudi.

**Methodology:** Laila Salameh, Amar Khamis, Syed Hammad Tirmazy, Youssef Dairi, Qutayba Hamid, Rifat Hamoudi.

**Project administration:** Laila Salameh, Bassam Mahboub, Rifat Hamoudi.

**Supervision:** Bassam Mahboub, Qutayba Hamid, Rifat Hamoudi, Saba Al Heialy.

**Writing – original draft:** Laila Salameh.

**Writing – review & editing:** Qutayba Hamid, Rifat Hamoudi, Saba Al Heialy.

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
