## [Decision Letter · Decision Letter 0]

16 Mar 2021

PONE-D-20-38174

Asthma Severity as a Contributing factor to Cancer Incidence: A Cohort Study

PLOS ONE

Dear Dr. Al Heialy,

Thank you for submitting your manuscript to PLOS ONE. After careful consideration, we feel that it has merit but does not fully meet PLOS ONE’s publication criteria as it currently stands. Therefore, we invite you to submit a revised version of the manuscript that addresses the points raised during the review process.

We look forward to receiving your revised manuscript.

Kind regards,

Maria Maddalena Sirufo

Academic Editor

PLOS ONE

Journal Requirements:

3. In the ethics statement in the manuscript and in the online submission form, please provide additional information about the patient records/samples used in your retrospective study, including the date range (month and year) during which patients' medical records/samples were accessed.

If patients provided informed written consent to have data from their medical records used in research, please include this information.

4. Please include a copy of Table 6 which you refer to in your text on page 11.

<h3>** **</h3>

5. We note you have included a table to which you do not refer in the text of your manuscript. Please ensure that you refer to Table 4 in your text; if accepted, production will need this reference to link the reader to the Table.

Reviewers' comments:

Reviewer's Responses to Questions

**Comments to the Author**

1. Is the manuscript technically sound, and do the data support the conclusions?

Reviewer #1: Yes

Reviewer #2: Yes

2. Has the statistical analysis been performed appropriately and rigorously? 

Reviewer #1: Yes

Reviewer #2: Yes

3. Have the authors made all data underlying the findings in their manuscript fully available?

Reviewer #1: Yes

Reviewer #2: Yes

4. Is the manuscript presented in an intelligible fashion and written in standard English?

Reviewer #1: Yes

Reviewer #2: Yes

5. Review Comments to the Author

Reviewer #1: This retrospective single-center cohort study evaluates the incidence of all cancer types in a population of asthma patients compared to a control population followed over a 9-year time interval. The role of asthma as a potential cancer risk factor is then evaluated.

This cohort study is well constructed. I would suggest a re-reading of the work in order to carry out a grammatical revision; for example on line 234: "the shows that ..."

The introduction investigates the relationship between cancer and chronic inflammatory diseases focusing on asthma. In the introduction we find: "Recent studies demonstrated that the association between history of allergic disorders and asthma, and the risk of cancer varied by cancer types". I would ask the authors to develop this part of the introduction more.

Reviewer #2: This paper is very well written. The results are very interesting for readers.The authors' conclusions are consistent with the data reported in the paper. I n conclusion the paper deserves to be published.

6. PLOS authors have the option to publish the peer review history of their article (what does this mean?). If published, this will include your full peer review and any attached files.

Reviewer #1: No

Reviewer #2: **Yes: **Urso Domenico Lorenzo

---

## [Author Response · Author response to Decision Letter 0]

31 Mar 2021

Journal Requirements:

3. In the ethics statement in the manuscript and in the online submission form, please provide additional information about the patient records/samples used in your retrospective study, including the date range (month and year) during which patients' medical records/samples were accessed.

If patients provided informed written consent to have data from their medical records used in research, please include this information.

We thank the editor for pointing this out. We have added the following paragraph in lines 85-89.

“All patient records were collected from the computerized information health system in Dubai Health Authority. All records in the outpatient and inpatient register, containing a hospital-discharge diagnosis of asthma. The data accessed were from 24/9/2019 – 30/4/2020. All patients were consented. Patient records were fully anonymized prior to access.”

4. Please include a copy of Table 6 which you refer to in your text on page 11.

We thank the editor for pointing out the anomaly. In fact, there is no Table 6. Table 4 was mislabeled as Table 6. This correction has been made in line 224. 

5. We note you have included a table to which you do not refer in the text of your manuscript. Please ensure that you refer to Table 4 in your text; if accepted, production will need this reference to link the reader to the Table.

Thank you for pointing out the anomaly. In fact, there is no Table 6. Table 4 was mislabeled at Table 6. This correction has been made in line 224. 

Reviewers' comments:

Reviewer's Responses to Questions

Comments to the Author

1. Is the manuscript technically sound, and do the data support the conclusions?

Reviewer #1: Yes

Reviewer #2: Yes

2. Has the statistical analysis been performed appropriately and rigorously?

Reviewer #1: Yes

Reviewer #2: Yes

3. Have the authors made all data underlying the findings in their manuscript fully available?

Reviewer #1: Yes

Reviewer #2: Yes

4. Is the manuscript presented in an intelligible fashion and written in standard English?

Reviewer #1: Yes

Reviewer #2: Yes

5. Review Comments to the Author

Reviewer #1: This retrospective single-center cohort study evaluates the incidence of all cancer types in a population of asthma patients compared to a control population followed over a 9-year time interval. The role of asthma as a potential cancer risk factor is then evaluated.

This cohort study is well constructed. I would suggest a re-reading of the work in order to carry out a grammatical revision; for example on line 234: "the shows that ..."

We thank the reviewer for taking the time to read through the manuscript. We have carefully gone through the manuscript and corrected the grammatical and other linguistic errors identified.

The introduction investigates the relationship between cancer and chronic inflammatory diseases focusing on asthma. In the introduction we find: "Recent studies demonstrated that the association between history of allergic disorders and asthma, and the risk of cancer varied by cancer types". I would ask the authors to develop this part of the introduction more.

We thank the reviewer for raising this point and agree that it is important to expand on it to introduce the fact that the relationship between the association of asthma and cancer. We have added the following statement in line 61-64:

“A protective effect of atopic diseases against pancreatic cancer has been shown consistently in case-control studies but not in cohort studies. Most studies on atopic diseases and non-Hodgkin lymphoma or colorectal cancer reported an inverse association.”

Moreover, this completes the statement in line 64-65:

“Inversion association between cancer and asthma have been reported in case-control studies of glioma [22, 23], and leukemia [24, 25].”

Reviewer #2: This paper is very well written. The results are very interesting for readers.The authors' conclusions are consistent with the data reported in the paper. I n conclusion the paper deserves to be published.

We thank the reviewer for their complimentary comments and for giving their time to go through the manuscript. We hope that the manuscript can add to the existing body of knowledge on the association between asthma and lung cancer.

6. PLOS authors have the option to publish the peer review history of their article (what does this mean?). If published, this will include your full peer review and any attached files.

Do you want your identity to be public for this peer review? For information about this choice, including consent withdrawal, please see our Privacy Policy.

Reviewer #1: No

Reviewer #2: Yes: Urso Domenico Lorenzo

---

## [Editor Report · Decision Letter 1]

7 Apr 2021

Asthma Severity as a Contributing factor to Cancer Incidence: A Cohort Study

PONE-D-20-38174R1

Dear Dr. Al Heialy,

We’re pleased to inform you that your manuscript has been judged scientifically suitable for publication and will be formally accepted for publication once it meets all outstanding technical requirements.

Kind regards,

Maria Maddalena Sirufo

Academic Editor

PLOS ONE
---

## [Editor Report · Acceptance letter]

16 Apr 2021

PONE-D-20-38174R1 

Asthma Severity as a Contributing factor to Cancer Incidence: A Cohort Study   

Dear Dr. Al Heialy:

I'm pleased to inform you that your manuscript has been deemed suitable for publication in PLOS ONE. Congratulations! Your manuscript is now with our production department. 

Kind regards, 

on behalf of

Dr. Maria Maddalena Sirufo 

Academic Editor

PLOS ONE